# Bacterial Nanocellulose in Dentistry: Perspectives and Challenges

**DOI:** 10.3390/molecules26010049

**Published:** 2020-12-24

**Authors:** Hélida Gomes de Oliveira Barud, Robson Rosa da Silva, Marco Antonio Costa Borges, Guillermo Raul Castro, Sidney José Lima Ribeiro, Hernane da Silva Barud

**Affiliations:** 1BioSmart Nanotechnology, LTDA, Av. Jorge Fernandes de São Mattos, 311, Incubadora Municipal de Araraquara, Araraquara, SP 14808-162, Brazil; biosmartnano@biosmartnano.com; 2Department of Chemistry and Chemical Engineering, Chalmers University of Technology, 412 96 Gothenburg, Sweden; rosa@chalmers.se; 3Laboratório de Biopolímeros e Biomateriais, University of Araraquara, UNIARA, Rua Carlos Gomes, 1217-Centro, Araraquara, SP 14801-340, Brazil; odontomacb@yahoo.com.br; 4UNLP Departamento de Química, Universidad Nacional de La Plata, Buenos Aires CP1900, Argentina; grcastro@quimica.unlp.edu.ar; 5Institute of Chemistry, São Paulo State University-Unesp-Rua Professor Francisco Degni, Araraquara, SP 14800-060, Brazil; sidney.jl.ribeiro@unesp.br

**Keywords:** bacterial cellulose, biomaterials, nanocomposites, tissue engineering, guided tissue regeneration

## Abstract

Bacterial cellulose (BC) is a natural polymer that has fascinating attributes, such as biocompatibility, low cost, and ease of processing, being considered a very interesting biomaterial due to its options for moldability and combination. Thus, BC-based compounds (for example, BC/collagen, BC/gelatin, BC/fibroin, BC/chitosan, etc.) have improved properties and/or functionality, allowing for various biomedical applications, such as artificial blood vessels and microvessels, artificial skin, and wounds dressing among others. Despite the wide applicability in biomedicine and tissue engineering, there is a lack of updated scientific reports on applications related to dentistry, since BC has great potential for this. It has been used mainly in the regeneration of periodontal tissue, surgical dressings, intraoral wounds, and also in the regeneration of pulp tissue. This review describes the properties and advantages of some BC studies focused on dental and oral applications, including the design of implants, scaffolds, and wound-dressing materials, as well as carriers for drug delivery in dentistry. Aligned to the current trends and biotechnology evolutions, BC-based nanocomposites offer a great field to be explored and other novel features can be expected in relation to oral and bone tissue repair in the near future.

## 1. Introduction

Biocompatible materials and devices have attracted a great deal of interest over the past years in the medical field. Ongoing efforts from many researchers are providing novel systems that mimic the main peculiar details of native tissues. Particularly, biocompatible materials engineered with improved functionalities and complex design have been proved to offer better cell adhesion, proliferation, and differentiation [1,2,3,4].

Generally, biomaterials are in direct contact with biological tissues and they can be defined as part of a system that deals with improving or replacing any tissue, organ, or body function [5]. Therefore, it is fundamental to carefully consider some requirements to appoint a material as a biomaterial, such as excellent biocompatibility and biodegradability and lack of toxicity.

According to its chemical nature, biomaterials can be broadly classified into two main categories: (I) Natural biomaterials, for example, collagen, silk, keratin, elastin, most polysaccharides, and tissues, such as bovine pericardium; and (II) synthetic biomaterials, which include ceramics, synthetic polymers, metals and alloys, and composites. Although their obtention is simple with a relatively low cost, they can cause some collateral effects.

Noteworthy, natural biomaterials directly help in regulation of the cellular phenotype [1,6,7] in addition to enhancement of tissue biocompatibility. As an example, collagen, a natural polymer, is the main organic compound of bone tissue, being widely explored in manufacturing biomaterials. However, its high cost, handicap in quality control related to contaminations, and standard commercial sources are challenging issues that hinder its use [8].

Naturally occurring biopolymers, viz. collagen, hyaluronan, gelatin, chitosan, and cellulose, are being explored in dentistry because their properties are like those of the native tissue. Several strategies were carried out to produce BC micro and nanocomposites for different biological and technological applications [9]. In particular, there is an emerging and appealing interest in bacterial cellulose towards its use as dental material [10,11]. In the pursuit to outline the known applicability and explore the ongoing efforts of BC in odontology, this review highlights in detail several up-do-date reports covering biomedical applications of BC in implants and scaffolds for tissue engineering, carriers for drug delivery, wound dressing, and applications related to dental materials as graphically summarized in Figure 1.

## 2. Bacterial Cellulose

Plant cellulose is the most abundant natural biopolymer on Earth, being harvested mainly from trees and cotton. Additionally, cellulose can also be produced by a wide variety of living species, such as microorganisms. Specifically, the production of cellulose from bacteria sources was first reported by Brown in 1886 [12]. Brown observed that the Gram-negative bacteria *Komagataeibacter xylinus* (previously named as *Gluconacetobacter xylinus*) was the only known species capable of producing cellulose on an industrial scale. Today, there are several Gram(-) bacteria able to produce BC, such as Agrobacterium, Pseudomonas, Rhizobium, and Escherichia spp. In an appropriate medium, these bacteria secrete an abundant 3-D network of cellulose fibrils under aerobic conditions, using glucose and other saccharides as a carbon source. The BC fibrils are condensed in a network located in the interphase of the culture media-air. The macroscopic aspect of the BC membrane is similar to a gel (Figure 2a), whose form and thickness depend on the recipient and cultivation time, respectively.

In terms of composition, BC has a structure very similar to plant cellulose but with superior physicochemical properties [13]. This feature is mainly addressed to the well-arranged 3-D network of fibers with diameters ranging from 3.0 to 3.5 μm (as shown in Figure 2b), which in turn are assembled by bundles of thinner cellulosic fibers with diameter sizes down to the micro- and nanoscale. Compared to plant cellulose, BC fibers are free of lignin and hemicellulose, contain small fibrils (100 times lower than plant cellulose), and have a highly crystalline structure [13,14,15,16]. Additionally, BC can be sterilized through many treatments, such as heat, steam, ethylene oxide gas, and radiation, without losing its intrinsic physicochemical properties and structural integrity.

BC fibrils are biosynthesized by polymerization of UDP-glucose into α-1,4-glucan chains by multienzymatic complex in several steps. BC fibers possess a strong tendency for self-assembly and form an extended network via both intramolecular and intermolecular hydrogen bonds due to their strong surface-interacting hydroxyl groups [17,18], enabling the production of sheets with a high surface area and porosity. Because of its uniform structure and morphology, the BC is endowed with peculiar characteristics, such as a high water retention capacity, high purity and crystallinity, good chemical stability, and remarkable mechanical properties [16]. Notwithstanding, BC unveils an exciting class of nanomaterial and an ideal starting point for useful biomaterials to be applied in therapeutics.

BC can be obtained from bacterial cultures in many formats and for different purposes, such as membranes under static culture conditions, and as micro- or nano-structures in shaken cultures. Alternatively, BC membranes can be processed to obtain microcrystals by physical (i.e., sonication) and/or chemical (i.e., sulfuric and nitric acids, periodate, etc.) treatment [19,20], or by enzymatic modification to produce nanofibrils [21].

With regards to physical, mechanical, and biological properties Czaja et al. [22] reported that BC could act as a physical barrier against microbial infections. Besides its low cost, BC is a biomaterial of feasible sterilization and manipulation, and is a non-allergenic and non-toxic material. In addition, the same authors reported that BC may significantly decrease pain when applied as a dressing material during certain topical treatments.

The literature describes several in vivo studies in which BC was applied, from the replacement of abdominal skin [23], dorsal implants [24], and brain and breast cancer treatments in rats [25,26], to vascular stents [15,27,28]. Other studies revealing the use of BC as artificial skin, artificial blood vessels and microvessels, wound dressing of second- or third-degree burn ulcers, and dental implants [23] were also reported.

Natural polymers and hydrogels (such as BC) possess many properties that make them attractive for challenging reconstruction problems, such as neural tissue applications for example [29]. Benefiting from high biocompatibility and bioadhesivity on living tissues BC has been widely used as a scaffold for cellular growth and tissue engineering [16,24,30,31]. The refined natural 3-D nanofibers of BC networks are distinct from the usual scaffolding due to the similarity of its shape to the collagen nanofibrils in natural tissue, such as the umbilical cord [30] and basement membrane in the cornea [32]. In addition, some researchers observed that endothelial, smooth muscle cells, and chondrocytes show good adhesion to BC [13,33].

To emphasize the relevance of the subject, over the past decade, several BC-based materials have been designed for a variety of biomedical applications, which has a marked notable increase in the number of scientific reports since 2000. The reader is advised to consult the cited review articles that highlight the potential applications of BC as a biomaterial [10,34,35,36,37,38,39,40,41] and recently in dentistry [42].

## 3. A Brief Overview of BC Uses in Biomedicine Applications

One of the first designed and main direct applications of BC membranes in the biomedical area is as a wound dressing in the replacement of burned skin [43]. Since then, the literature has shown an increasing number of papers related to wound dressing. In terms of a temporary covering, BC dressings are mightily recommended by manufacturers for the treatment of different types of wounds, including skin tears, pressure sores, venous stasis, second-degree burns, ischemic and diabetic wounds, traumatic abrasions and lacerations, and skin graft and biopsy sites [44].

Currently, it is possible to identify the following BC-based wound dressings available on the market: Gengiflex^®^ (for periodontal reconstruction), XCell^®^, Bioprocess^®^, and BioFill^®^ [45,46]. Within the mentioned brands, BioFill^®^ represents one of the first commercial products that fulfills the main requirements of an ideal wound dressing, including: water vapor permeability, good adherence to the wound, transparency, elasticity, durability, a physical barrier for bacteria, hemostatic, easy handling, and low cost with minimum exchanges. In addition, BioFill^®^’s efficiency in accelerating the healing and pain relief process has been successfully proven in more than 300 clinical trials in humans [22,45,46]. Additionally, it has been reported that the BC wound dressing clearly shortened wound closure and/or the healing time over standard care when applied to non-healing lower extremity ulcers in humans [22,47,48].

Likewise, Portal and co-workers [47] also observed the healing process by using BC dressing (DermafillTM, AMD/Ritmed, Tonawanda, NY, USA) for chronic wounds. They found a considerable reduction in wound closure time from 315 days to 81 days using a BC membrane, with 75% of epithelialization of the affected areas.

Despite the fact that the analgesic mechanism of action and the sensation of pain relief of these wound dressings is not yet fully understood, some authors suggested that the healing could be related to the high number of hydroxylic groups. These hydroxylic groups are able to retain a huge amount of water concomitantly with the capture of ions by means of cellulose hydrogen bonds or by the nano BC 3-D network, which mimics the skin surface, resulting in optimal conditions for healing and regeneration [22,46].

A variety of surface functionalization’s through biosynthetic or chemical modification was also investigated by incorporating different substrates, including polymers, inorganic nanoparticles or nanowires, and small molecules, on the surface of BC that can be easily maneuvered, forming nanocomposites with fine-tuned properties. Therefore, several BC-based nanocomposites have been developed with improved mechanical and mucoadhesive properties, such as BC/collagen [49,50,51,52] and BC/gelatin [53,54].

Kim et al. [55] prepared wet BC-gelatin nanocomposites, and NIH3T3 fibroblast cells were further seeded and compared with pure BC. After 48 h of incubation, they found that the cells spread well, presenting good adhesion in both platforms, although the biocompatibility was more superior in BC-gelatin composites than that of pure BC.

Besides collagen and gelatin protein, nanocomposites of BC modified with naturally occurring biopolymers, such as aloe vera, have been tailored as biomaterials for therapeutic applications [56,57].

Lin et al. [58] reported the skin wound healing efficacy of BC membranes modified with chitosan in experiments assessed with rat models. They found that the resulting nanocomposite reduced the time for wound healing and did not produce any toxic effect on animal cells.

BC/silk fibroin (SF) sponge-like scaffolds were obtained by Oliveira Barud et al. [40] by means of the freeze-drying technique. These nanocomposites presented a non-cytotoxic or genotoxic character as is possible to observe in SEM images, which showed a greater number of L929 cells attached to the BC/SF: 50% scaffold surface when compared with pure BC. The huge difference in terms of cell attachment may suggest that the presence of fibroin improved and created optimal conditions. So, the SF amino acid sequence may act as cell receptors guiding and allowing easy cell adhesion and growth (Figure 3), pointing that BC/SF: 50% scaffolds represent an excellent option in bioengineering and tissue regeneration in the cultivation of cells on nanobiocomposites.

Other composites, such as BC-chitosan, BC-poly(ethylene glycol), and BC-collagen, revealed improved NIH3T3 cell activity as compared to native BC [50,59]. On the other hand, human adipose-derived stem cells proliferated on the BC-poly(2-hydroxyethyl methacrylate) scaffold to a lower extent in comparison to pure BC membranes [60]. This fact could be attributed to the presence of monomer residues present in the hybrid scaffold structure as well as the partial biocompatibility of this compound since it is considered an allergen [9].

Nanocomposites with antimicrobial activity have been prepared by incorporating silver nanoparticles [14,61,62]. Furthermore, BC nanocomposites modified with propolis, a naturally occurring substance, have garnered a great deal of attention as biomaterials due to its remarkable antifungal, antiviral, antioxidant, anti-inflammatory, and antibacterial properties [63].

In recent years, several controlled release systems based on nanocellulose have been conceived. Noteworthy, BC fibrils were used to deliver a myriad of drugs: Tetracycline [64], benzalkonium chloride [65], topical release of lidocaine and ibuprofen [66,67], doxorubicin [9], and release of proteins with serum albumin [68].

Focusing on vascular applications, a German group created BASYC^®^ (Bacterial Cellulose Synthetized), a tubular biomaterial to be used in microsurgery of arteries and veins [27,28]. Similarly, Bodin et al. [69] modified the fermentation process of *Komagataeibacter xylinus* using silicone guides to mold and obtain BC tubes.

Nimeskern et al. [70] designed the biofabrication of a patient-specific ear-shaped BC implant as a prototype device by applying a 3-D bioprinter technique, which consists of a high-precision motion and a microdispensing system to construct the device layer by layer. The application of this modern and innovative technique confirms BC’s great potential in bioengineering and as an ear cartilage replacement material with appropriate mechanical properties.

Bent BC films have been fashioned by using a special machine in order to shape the biocompatible wound dressing contact lens, seeking their use in the regeneration of cornea tissue or as drug delivery systems [71].

Recently, an implant for a cell trap made of BC after glioblastoma brain surgery was reported [25]. The work suggested that BC could concentrate and trap cancer cells without impact on brain parenchyma and the visibility of BC scaffolds by magnetic resonance imaging allowed increased precision of the stereotactic radiosurgery.

A plentiful number of works also unveil other interesting BC applications in tissue engineering related to neural implants/dura máter [72,73,74], urinary conduits [75,76], tympanic membrane [77,78], and vocal folds [79].

## 4. BC in Dentistry

Despite the great applicability in biomedicine and tissue engineering, BC-based materials are still poorly explored in dentistry.

There are few studies that evidence the applicability of BC in dental or oral fields. Some of the first applications in the dental area were related to the guided tissue regeneration technique (GTR), aiming at periodontal disease treatment. The biologic basis of GTR to promote periodontal regeneration is related to the placement of physical barriers into specific sites, which will prevent apical migration of the epithelium and gingival connective tissue cells and will allow an isolated area for the migration of mesenchymal and periodontal ligament cells (PDL) over the exposed root surface. The role of physical barriers goes far beyond aesthetic appeal: besides further selective repopulation of the affected area by the maintenance for ingrowth of a new periodontal tissue, they provide protection of the blood clot during early cicatrization stages. However, GTR membranes have limited clinical efficacy on biologic effects related to not providing differentiation and the proliferation of mesenchymal and PDL cells [80,81].

Among the lesions treated by GTR are class II furcation lesions, two-to-three wall infrabone defects, extensive defects combined or not with other techniques, bone dehiscence, and lesions associated with implants [82,83].

Novaes Jr et al. [84] treated class II furcation lesions in humans using a non-resorbable BC membrane (Gengiflex^®^) with the GTR technique. Clinical evaluation of attachment levels, radiographs, and a re-entry process were applied, and it was possible to observe the closure of the defect. In [85], the same group successfully treated class II furcation lesions in dogs with naturally occurring periodontitis using the same membrane.

Adequate GTR results in periodontal defects in humans as well as in GTR for bone formation were reported by Novaes [86,87] using the commercial membrane Gengiflex^®^. The mentioned membrane was successfully applied in association with bone-integrated implants.

An et al. [88] prepared bacterial cellulose membranes (BCMs) and performed tests in rats to assess guided bone regeneration. The electron beam (EI) irradiation technique was used to increase the biodegradation of BC. Thus, EI-BCMs membranes were evaluated by Fourier transform infrared attenuated spectroscopy (ATR-FTIR), scanning electron microscopy (SEM), thermal gravimetric analysis (TGA), and other measures of wet tensile strength as well as in vitro analysis to confirm cytocompatibility. All these chemical, mechanical, and biological analyses showed effective EI–BCMs interactions with cells, which promoted bone regeneration as a result.

On the other hand, surgical wounds of the oral mucosa are important issues in dentistry, as shown in Figure 4. Small extent defects usually heal on primary closure, but split or full-thickness grafts can be used for moderate ones. For defects involving most of the buccal mucosa, there is a need for a second surgical area [83].

Regarding tissue reconstruction, there are several key remarkable points. One of the most relevant considerations is minimizing flap donor site morbidity, as it is known that the gold standard of a graft material is the autogenous one. Some substitutes based on biomaterials have been developed due to the following issues: morbidity of the donor site, the risk of postoperative complications, the time of the procedure surgical, the unpredictable resorption rate, and the limited amounts of soft tissue available [83].

Chiaoprakobkij et al. [89] prepared a novel three-dimensional composite based on bacterial cellulose/alginate (BCA) to be used as a temporary dressing of oral surgical flaps. In this study, HaCat cells (a keratinocyte cell line) were seeded on BC and BCA scaffolds and compared to gingival fibroblast cells. It was observed that the last-mentioned cell line had attached only onto the BCA sponge, which led researchers to conclude that the BCA sponge scaffold has good potential for use in the oral cavity to cover surgical wounds. Figure 4 illustrates some examples of the BC-based temporary wound dressing. The material features a unique design: the outer layer is thicker to prevent bacterial contamination and dehydration of the wound whereas the inner layer is porous and designed to drain exudates.

With respect to bone regeneration in dentistry as well in biomedicine, the autogenous bone graft is still considered the gold standard. Although it is the only one that presents osteoinductive, osteoconductive, and osteoprogenitor properties [90,91,92], it presents the same risks and morbidity mentioned above. Furthermore, in terms of form and function, each anatomical area offers unique challenges for bone reconstruction, and some graft donor sites (like the fibula, scapula, radio, and iliac crest) can lead to increased morbidity of the donor site [93]. As alternatives to regenerate bone tissue, allogeneic [94,95], xenogenous [96], or synthetic biomaterials have been evaluated as grafts [97,98], as shown in Figure 5.

At the current stage of development, the efforts of the most recent research seem to be focused on the development of synthetic bone substitutes that are able to mimic extracellular matrix functions of natural tissues and lead the host response, offering similar results to the autograft implants [98,99].

BC has been recently exploited to design a tailorable matrix to synthesize different types of calcium carbonate (CaCO_3_) and hydroxyapatite crystals from different starting reagents with improved biocompatibility. For instance, in order to synthesize CaCO_3_ over BC membranes, Stoica-Guzun et al. [100] used sodium carbonate (Na_2_CO_3_) and calcium chloride (CaCl_2_) as starting reactants. Other authors [101,102] produced BC-hydroxyapatite (Hap) nanocomposites by an optimal biomimetic mineralization route, inducing a negative charge on BC nanofibrils, which stimulated the nucleation of Hap via simulated body fluid (SBF) by the adsorption of polyvinylpyrrolidone (PVP). Before starting the biomimetic mineralization process, Shi et al. [103] developed an alkaline treatment in order to improve the mineralization efficiency and results. Further, Zhang et al. [104,105] promoted the growth of Hap by using a phosphorylation reaction to introduce phosphate groups to BC’s surface.

Dental implants are routine procedures in clinical practice, and often, the upper jaw region has insufficient bone height for the procedure. Thus, Boyne and James [106] created the maxillary sinus lifting technique by performing a graft, allowing implant insertion. In this way, Koike et al. [107] performed frontal bone defects in 12 rabbits that were divided in 4 groups: BC (BC grafting only), BMP-2 (treated only with BMP-2 solution), and BC + BMP-2 (BC loaded with BMP-2 graft). As a result, BC maintained the graft space and BMP-2 was released in a controlled manner into the target area, showing that BC + BMP-2 is a promising option to increase bone structure and for the placement of dental implants.

Saska et al. [108] evaluated the performance and biological properties of BC-Hap nanocomposites in defects of the rat tibiae, aiming at bone repair. The BC-Hap membranes were effective in inducing new bone formation as a slow reabsorption of the membranes occurred, suggesting that longer periods are needed for this compound to be fully reabsorbed. Similarly, Tazi et al. [109] investigated BC-Hap nanocomposite for potential bone regeneration. As a result, BC-Hap nanocomposites stimulated the growth of osteoblast cells, with a high level of alkaline phosphatase activity, and which resulted in greater sites of bone formation. The better the osteoblasts’ adhesion, the better BC-Hap biomaterials are expected to offer cell proliferation and mineralization to encourage faster bone tissue regeneration. Grande et al. [105] also developed BC-Hap scaffolds, presenting excellent results concerning bone and connective tissue regeneration.

Coelho et al. [110] synthesized an innovative BC membrane associated with HA and an antibody of a bone morphogenetic protein (anti-BMP-2) (BC-HA-anti-BMP-2). The SEM-EDS and FTIR assays confirmed the presence of BC and HA. The proposed biomaterial increased the expression of specific genes related to bone repair, proving to be non-cytotoxic, genotoxic, nor mutagenic biomaterial by using MC3T3-E1 cells.

In an attempt to manufacture new biomaterials to induce bone repair, Fan and coworkers [111] reported the introduction of goat bone apatite into BC. The designed biomaterial demonstrated in vitro cell differentiation and also stimulated cell adhesion and proliferation.

In reference to critical-sized calvarial defects in mice, Pigossi et al. [112] evaluated the performance of BC-Hap composites associated with osteogenic growth peptide (OGP) or pentapeptide OGP (10–14) as a potential biomaterial in bone regeneration. Thus, BC-Hap-OGP, BC-Hap-OGP (10–14), and BC-Hap membranes were analyzed at 3, 7, 15, 30, 60, and 90 days. Within each period of analysis, all specimens were evaluated by descriptive histology, micro-computed tomography (µCT), VEGFR-2 (vascular endothelial growth factor) quantification by ELISA, and gene expression of bone biomarkers by qPCR. The researchers found that BC-Hap-OGP (10–14) and BC-Hap membranes reveled at 60 and 90 days a high percentage of bone formation by µCT. High expression of some bone biomarkers, such as Tnfrsf11b and Alpl, Spp1, were also noticed, which lead them to conclude that the BC-Hap membrane promoted better results related to bone formation in critical-sized mice calvarial defects.

In terms of bone repair, Lee and coworkers [113] evaluated silk fibroin-BC membranes’ performance by successfully implanting them on bilateral segmental defects (2 mm in length) of rats’ zygomatic arches. Complete healing of the surgical wounds was observed at the 8-week follow-up against bone degeneration and necrosis when compared to the control side without fixation.

Concerning the field of endodontics, Figure 6 summarize some BC applications. BC has been reported as an innovative material for dental root canal treatment in animal experiments by Yoshino et al. [114]. In comparison with commercial paper point materials (ISO 45 JM paper point, Morita, Osaka, Japan), BC did not exhibit harmful effects, presenting safe compatibility and biological characteristics for dental root canal treatment. Furthermore, BC-based biomaterials under wet conditions showed an excellent absorption rate without deformation in relation to conventional ones.

After the cleaning and drying steps related to endodontic treatment, the ideal sequence concerns a complete sealing of the root canal space with filling materials.

Due to the excellent properties of BC, it can enable the production of whiskers in the nanometric scale (nanowhiskers) that can be used as a reinforcement for other materials, including dental cements. Thus, Jinga et al. [115] prepared BC nanowhiskers and used commercial MTA (mineral trioxide aggregates) cement for the preparation of some composites: MTA-E (mineral trioxide aggregates-experimental), composites MTA-10% biocell, and MTA-33% biocell cements. Considering one day of hardening, XRD patterns, and the thermal analysis data, they concluded that the presence of BC nanowhiskers accelerates the hardening processes of MTA cement while a decrease in the amount of calcium hydroxide crystals was noted. Thus, it was firstly reported in the literature that BC nanowhiskers further the formation of crystalline hydrosilicates even after one day of hardening, without affecting the viability and affinity of cells.

Intending to promote a tissue regeneration response by applying the dental pulp capping technique, Manzine Costa and coworkers [116] developed an otolith/BC nanocomposite (OTL). Subsequently, exposed incisor dental pulps of dogs were divided into two groups: BC/Otolith preparation (OTL) and no protective material in the control group. The teeth were then sealed with glass ionomer and histological analyses were performed after 21 days. The proposed OTL nanocomposite induced the formation of a tissue barrier expressing osteoid-like characteristics of mineralization. The obtained results indicate that otolith/cellulose bacterial nanocomposites might work as a potential dental pulp capping biomaterial.

Recently, Voeicu et al. [117] synthesized a new family of composites starting from mineral binder powders and biocellulose membranes, showing great applications in endodontics. Silicate cement synthesized through the sol-gel technique was prepared by the addition of a BC polygranular powder previously obtained (crushed into micrometric particles via hydrothermal treatment). The resulting nanocomposite was investigated by SEM, X-ray diffraction, thermal analysis, and other mechanical tests. The most important effect of the BC content was observed in terms of shortening the setting time, where reduced clinical care time is required. In vitro tests were performed, and all materials did not exert cytotoxic effects, and promoted cell adhesion and proliferation. Moreover, they presented a pronounced mineralization process in the simulated conditions. The authors concluded that the proposed composite has high potential for applications in endodontic treatments, mainly in root canal filling, root perforations, or dentin remineralization.

Regarding the reinforcement properties, the BC presented good results, improving the mechanical properties when added to chitin fibers in surgical suture applications, and thus setting up a promising new candidate as a BC-based medical suture for dentistry [118].

Carvalho et al. [119], in 2020, developed BNC-based patches containing both HA and (Diclofenac) DCF. The objective of the material was the stimulation of healing of the aphthous ulcers in recurrent aphthous stomatitis (RAS) and the mitigation of pain. RAS is the most common form of oral mucosa ulcer, affecting from 5% to 66% of the world’s population, which is also known as aphthae or canker sores. Despite manifesting spontaneous healing in a few days, they can be extremely uncomfortable, causing stinging pain and local inflammation. From this perspective, the freestanding membrane patches were fabricated via simple diffusion of HA and DCF aqueous solutions into a wet BNC three-dimensional porous network. The resultant nanostructured patches were thermal resistant and stable up to 200 °C. In vitro assays showed that the patches were almost 100% after 24 h of incubation. In addition, the swelling ability and DCF release was conducted in simulated saliva, pointing to controlled drug-release purposes. The attained results hint at the possibility of using the proposed BNC-HA-DCF-based patches to diminish and treat aphthous stomatitis discomfort in the oral mucosa.

Finally, Table 1 shows a brief summary of the main BC advantages concerning its properties to be applied in dental and oral fields.

## 5. Conclusions

As briefly shown, bacterial cellulose presents excellent properties that makes it a wonderful renewable polymer synthesized from the bacterium *Komagataeibacter xylinus* or related bacteria, which has attracted considerable interest and applications in several biomedical and tissue engineering fields. Although BC’s advantages and properties are not new, this review showed that BC can be explored as an innovative biomaterial for dental and oral applications.

Probably, it is possible to observe an increasing degree of complexity in BC dental applications. Due to some BC properties, such as transparency, elasticity, durability, acting as a physical barrier against bacteria, hemostatic, easy handling, low cost, accelerated healing process, and pain relief, firstly, pure BC membranes were directly applied to regenerate the oral mucosa and periodontal tissue. Other properties, such as easy moldability and natural 3-D well-arranged nanofibers, have inspired new BC-based nanocomposites to perform dental endodontic treatment, surgical, reinforcement biomaterial, scaffolds, and nanocomposites aimed at bone mineralization, for example.

There is huge potential to be explored, mainly related to tissues that have limitations or difficulties in the healing process, such as nerves, cartilage, and bone. Although some BC-based materials are focused on bone tissue, bone regeneration is a complex physiological process that needs time for its formation. Bacterial cellulose is a good candidate for bone repair as it presents versatility for the preparation of nanocomposites by a biomimetic approach. Aligned to the current trends and taking advantage of biotechnology and the high-value-added functional materials, many other novel features can be expected in the near future, especially those related to 3-D nanomaterials and scaffold design. Thus, the material properties can be enhanced when associated with a biomaterial because in addition to promoting adhesion and proliferation of target cell lines, they can still act as potential candidates in other applications associated with bone regeneration, like alveolar bone regeneration and fixation, cleft palate, and dental implants.

## Figures and Tables

**Figure 1 molecules-26-00049-f001:**
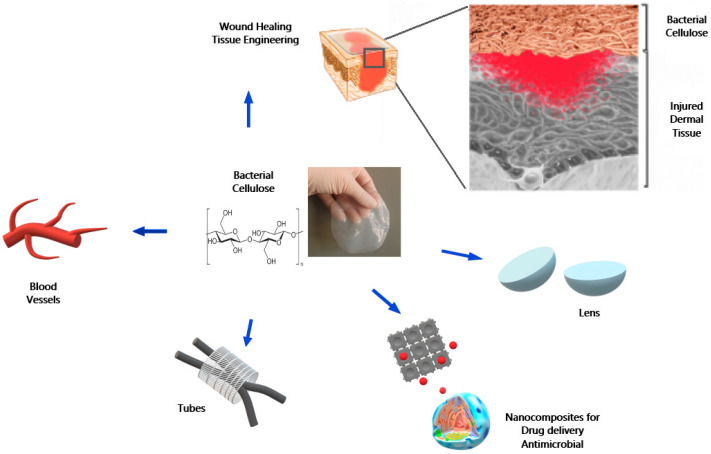
Some examples of main applications of BC in biomedicine.

**Figure 2 molecules-26-00049-f002:**
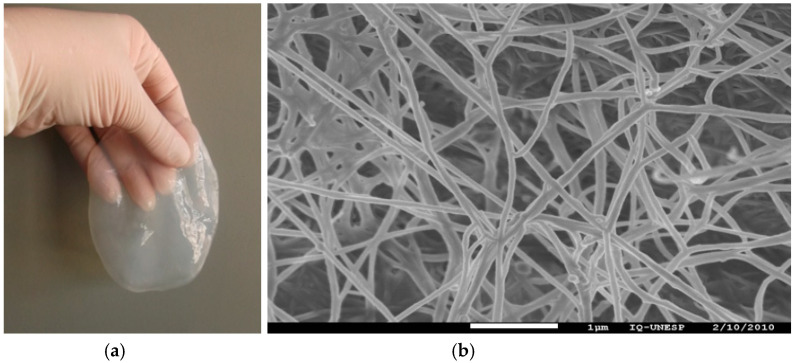
(**a**) The gel-like macroscopic aspect of a BC membrane with 4 days of cultivation and (**b**) top-view SEM image of the BC fibers network.

**Figure 3 molecules-26-00049-f003:**
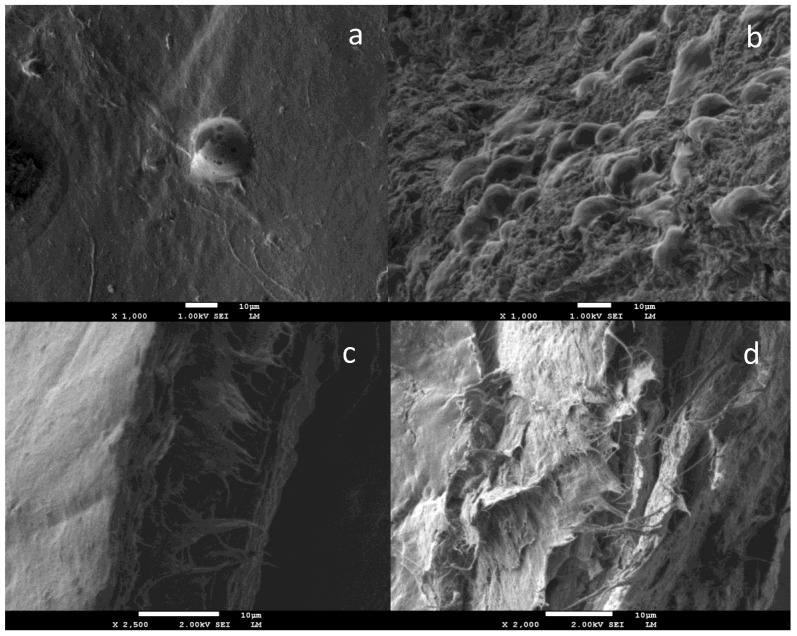
SEM images of L-929 cells attached to BC (**a**) and BC/SF (**b**) scaffolds surface at 48 h; cross-section SEM images of BC (**c**) and BC/SF (**d**) evidenced that the cells did not migrate into the scaffolds. Reprinted with permission from [40].

**Figure 4 molecules-26-00049-f004:**
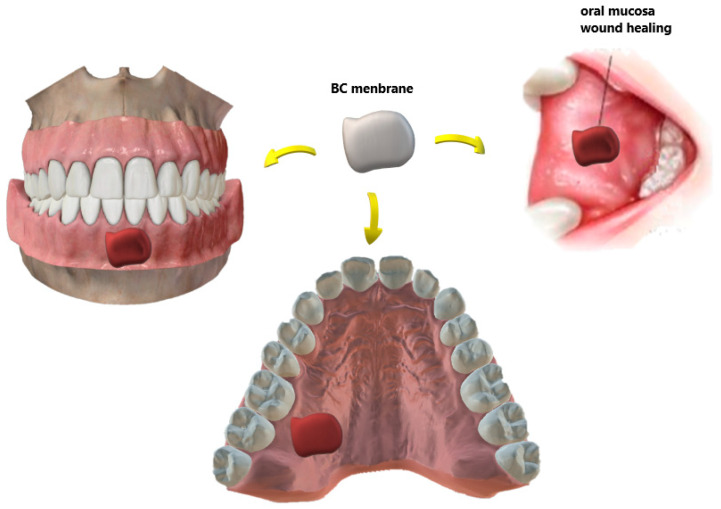
Different types of wounds in the oral mucosa recovered with biomaterials for BC-based dressings. In the case of an autograft donor site, the membrane minimizes the pain and morbidity of the site.

**Figure 5 molecules-26-00049-f005:**
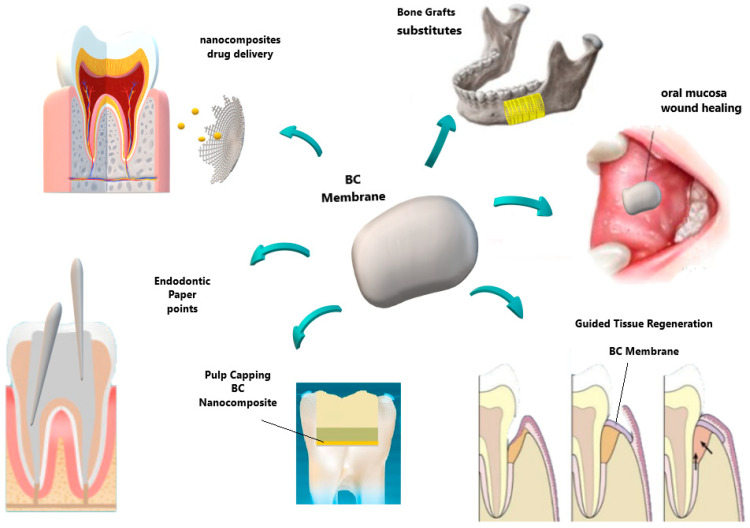
Schematic representation of some distinguished bacterial cellulose applications in dentistry, including the perspective of bone grafts substitutes.

**Figure 6 molecules-26-00049-f006:**
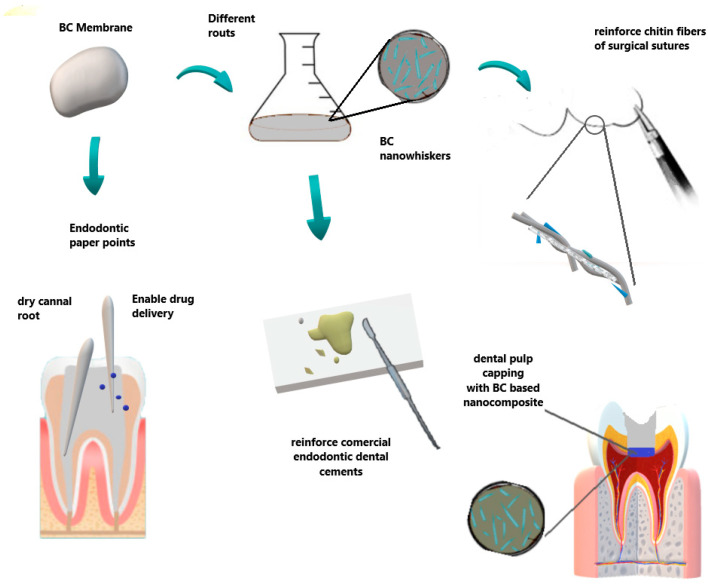
Some applications in endodontic and surgical areas. BC can be used as paper points to dry the root canal, also allowing desired target drug delivery. Otherwise, BC nanowhiskers can be obtained by different routes to reinforce dental cements and sutures in surgical procedures.

**Table 1 molecules-26-00049-t001:** Summary of the main BC advantages to be to be applied in dental and oral fields.

Dental and Oral Treatments	Potential Use	BC Advantages
Periodontal treatment	Barrier membrane in GTR technique Novaes Jr et al., [84,85] Novaes [86,87] An et al., [88]	Allowed cell attachment and proliferation Aesthetics importance restoration of oral function Reduction in surgical steps Biocompatibility, presenting no chronic inflammatory reaction
Wound dressing/patches	Surgical Wounds, flaps and RAS ulcers Chiaoprakobkij et al., [89] Carvalho et al., [119]	Biocompatible Physical barrier Allow drug delivery
Dental pulp tissue treatment	Root canal sealers Jinga et al., [115] Scaffold for regenerative Endodontic treatment Manzine Costa et al., [116] Voeicu et al., [117]	Accelerates the hardening processes of cement Reinforce dental cements Mimics extracellular matrix. Induces mineralized barrier and apical closure
Dental surgery	Surgical suture Wu et al., [118]	Improve reinforcement and mechanical properties

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
