# Peer review of "Bacterial Nanocellulose in Dentistry: Perspectives and Challenges"

_molecules, 2020, doi:10.3390/molecules26010049_

Round 1
Reviewer 1 Report
While the manuscript title suggests that authors want to present review of bacterial nanocellulose in dentistry, in the abstract they claim to present systematic analysis of recent literature about the biomedical use of bacterial cellulose (BC) on the design of implants, scaffolds and wound-dressing materials as well as carriers for drug delivery in dentistry. In my opinion the review did not meet the aim of the authors.
The section about BC in dentistry comes only in the second half of the manuscript, while the rest of the manuscript contains text that can only be described as a list of what other authors did, presented in few lines.
The review contains five figures, i.e. two schematics, one figure about publication statistics, and only two figures containing experimental data. I feel that that is not adequate for a paper that claims to be a review.
It would be much better if the authors could focus their presentation on each one of the aspects they want to talk about (design of implants, scaffolds, would-dressing, drug delivery in dentistry), provide broader picture of what is involved in each one of them and the role of BC in solving the problem.
I am grateful to the authors for aiming to provide this review, which contains lot of references, and can act as encouragement for further understanding of the subject, but I am of the opinion that the manuscript needs to be much improved before its publication.
Author Response
Molecules special issue "Cellulose Nanomaterials: Production and Applications"
Title: BACTERIAL NANOCELLULOSE IN DENTISTRY: PERSPECTIVES
AND CHALLENGES
Dear Editor and Reviewer,
Thank you for your useful comments and suggestions on the language and
structure of our manuscript. We have modified the manuscript accordingly, and detailed
corrections are listed below point by point:
#REVIEWER 1:
1. While the manuscript title suggests that authors want to present review of
bacterial nanocellulose in dentistry, in the abstract they claim to present systematic analysis of recent literature about the biomedical use of bacterial cellulose (BC) on the design of implants, scaffolds and wound-dressing materials as well as carriers for drug delivery in dentistry. In my opinion the review did not meet the aim of the authors.
The section about BC in dentistry comes only in the second half of the manuscript,
while the rest of the manuscript contains text that can only be described as a list of what other authors did, presented in few lines.
The intention of the manuscript was to provide quality information that can be accessed by researchers who need to be up to date on the biotechnological evolution that bio and nanomaterials are going through. Often the focus of the research is more concentrated on the development of materials with properties increasingly superior to those previously proposed. However, in terms of applications, the clinical reality in practice is still far from the advances already made in other areas of biomedicine, chemistry, physics. Unfortunately, many dentistry researchers working in the development of dental materials
have little or no knowledge about bacterial cellulose and its applications in biomedicine, and even less in dentistry. As the tendency is to work more and more in multidisciplinary teams, the idea of the article is to provide updated and easily accessible information to support future advances in dental practice.
2. The review contains five figures, i.e. two schematics, one figure about publication statistics, and only two figures containing experimental data. I feel that that is not adequate for a paper that claims to be a review.
It would be much better if the authors could focus their presentation on each one of the aspects they want to talk about (design of implants, scaffolds, would-dressing, drug delivery in dentistry), provide broader picture of what is involved in each one of them and the role of BC in solving the problem.
The structure of the manuscript was reviewed and now it presents 6 Figures and 1 table content. According to suggest of the other review, the figure related to the statistical graphics were removed. In relation to the suggestion of separating the figures by topic was partially accepted, since there was not enough time to reorder all topics.
Thus:
Figure 1, 2 and 3 are the same.
Figure 4: is directed to oral and mucosa dressings (it is new)
Figure 5: related to applications in dentistry (has been remodeled)
Figure 6: highlighted endodontic and material reinforcement aspects (it is new)
Table 1: summarizes the main advantages of using BC based biomaterials in dentistry
(new added content).
3. I am grateful to the authors for aiming to provide this review, which contains lot
of references, and can act as encouragement for further understanding of the
subject, but I am of the opinion that the manuscript needs to be much improved
before its publication.
We worked hard in short deadline time offered to answer all raised points. The manuscript was completely revised and some part of it was rewritten. We hope that the changes will be in accordance with the suggested. We appreciate all your suggestions that were of great help in improving the quality of the review article related to BC in dentistry.
The manuscript has been resubmitted to your journal. We look forward to your positive response.
Sincerely,
Dr. Hernane Barud
Reviewer 2 Report
This review shows a great effort to summarize the novelties related to the use of BC in dentistry.
Major changes:
- Unlike the abstract section, where information provided to the reader is clear and useful, the conclusion section just shows a good of intentions. This sections must be rewritten and meaningful ideals should be included.
- Figure 3 and paragraph lines 115-120 are not necessary. There is not relevant information in this part. Authors should express BC importance in biomedical application in other way.
Minor changes:
- Line 56: "Bacterial cellulose (BC)". However, the acronym has been used previously.
- Line 155: "48 h"
- Line 157: Not understood. Rewritte.
- Line 166 and Figure 3: L9292 an L-929.
- Line 181, 185: have/has been
- Line 264: Not necessary, delete or rewritte.
- "et al." is sometimes written in italics, but not always.
Author Response
Molecules special issue "Cellulose Nanomaterials: Production and Applications"
Title: BACTERIAL NANOCELLULOSE IN DENTISTRY: PERSPECTIVES
AND CHALLENGES
Dear Editor and Reviewer,
Thank you for your useful comments and suggestions on the language and
structure of our manuscript. We have modified the manuscript accordingly, and detailed corrections are listed below point by point:
#REVIEWER 2:
Major changes:
1- Unlike the abstract section, where information provided to the reader is clear and useful, the conclusion section just shows a good of intentions. This sections must be rewritten and meaningful ideals should be included.
2- Figure 3 and paragraph lines 115-120 are not necessary. There is not relevant
information in this part. Authors should express BC importance in biomedical
application in other way.
The points raised in each topic were all reviewed and rewritten, including the abstract, section 2, 3 and conclusion. The structure of the manuscript has been reviewed and now presents 6 figures and 1 table of contents. The suggestion to remove the old figure 3 was accepted and the paragraph was rewritten.
Thus:
Figure 1, 2 and 3 are the same.
Figure 4: is directed to oral and mucosa dressings (it is new)
Figure 5: related to applications in dentistry (has been remodeled)
Figure 6: highlighted endodontic and material reinforcement aspects (it is new)
Table 1: summarizes the main advantages of using BC based biomaterials in dentistry
(new added content).
Minor changes:
Line 56: "Bacterial cellulose (BC)". However, the acronym has been used
previously.
Line 155: "48 h"
Line 157: Not understood. Rewritte.
That sentence was a part of the text that was changed in the authors' corrections
and by mistake was not removed.
Line 166 and Figure 3: L9292 an L-929.
Line 181, 185: have/has been
Line 264: Not necessary, delete or rewritte.
"et al." is sometimes written in italics, but not always.
All of these suggestions and errors were considered and corrected, including
typographical ones. Thank you for all observations and suggestions. We hope that all positive changes were in accordance to expected.
The manuscript has been resubmitted to your journal. We look forward to your positive
response.
Sincerely,
Dr. Hernane Barud